# Sub-Synchronous Oscillation Suppression Strategy Based on Impedance Modeling by Attaching Virtual Resistance Controllers for Doubly-Fed Induction Generator

**Yingming Liu [1], Guoxian Guo [1,*], Xiaodong Wang [1], Hanbo Wang [1] and Liming Wang [2]**

[1]  Institute of Electrical Engineering, Shenyang University of Technology, Shenyang 110870, China; yingming.liu@hrwind.com (Y.L.); xiaodong.wang@hrwind.com (X.W.); wanghb@smail.sut.edu.cn (H.W.)

[2]  Institute of Electrical Engineering, Shenyang Institute of Engineering, Shenyang 110136, China; wanglm@sie.edu.cn

*  Correspondence: bccs123gx@smail.sut.edu.cn; Tel.: +86-024-25670358

**Abstract:** A sub-synchronous oscillation (SSO) suppression strategy of attaching virtual resistance controllers to the rotor-side converter (RSC) of the doubly-fed induction generator (DFIG) is proposed in this study to suppress sub-synchronous oscillation (SSO) caused by series compensation and grid connection of DFIG. A DFIG-based frequency domain impedance model considering RSC control under small signal perturbations is developed in a three-phase stationary coordinate system. Subsequently, the factors and mechanisms of SSO in the system with different phase sequences are analyzed in combination with the equivalent RLC resonant circuit of a DFIG-based series-compensated grid-connected system (SCGCS). SSO occurs when RSC and rotor winding generate a large equivalent negative resistance at the SSO frequency, resulting in a negative total system resistance. Additionally, the influences of series compensation degree (SCD) of line and inner loop parameters (ILPs) of RSC related to the total impedance of the system on the SSO characteristics are analyzed to optimize the parameters and improve the system stability. Based on the causes of SSO, virtual resistance controllers are attached to RSC to provide positive resistance to the system and to offset the equivalent negative resistance of RSC and rotor winding at the SSO frequency, thereby avoiding SSO of the system. Finally, time-domain simulations using power system computer aided design/electromagnetic transients including dc (PSCAD/EMTDC) show that the SSO of the system is effectively suppressed.

**Keywords:** doubly-fed induction generator (DFIG); sub-synchronous oscillation (SSO); rotor-side converter (RSC); frequency domain impedance; virtual resistance

## 1. Introduction

The global emphasis on low-carbon and environmental protection in recent years has led to new energy power generation gradually becoming a research hotspot. In the Three-Northern Region of China (including north-eastern, northern, and north-western China), wind resources are abundant, which provides a favorable foundation for vigorously utilizing wind energy, and wind power generation has been rapidly promoted. Due to the long distance between most wind farms in the Three-Northern Region and the eastern developed cities, long-distance high-voltage transmission is necessary. At present, series compensation capacitors are generally used for the transmission of electrical energy to improve the transmission capability of the line. However, series-compensated transmission has the problem of inducing sub-synchronous oscillation (SSO) in the system, which reduces the working safety of the wind turbine and threatens the stable operation of the power system [1–4]. Therefore, studying the SSO suppression strategy can improve the resilience of the DFIG itself and the stability of the whole power system.

Currently, the main methods for conducting SSO-related studies include the frequency sweep method [5,6], the time-domain simulation method [7–9], the eigenvalue analysis

method [10,11], and the impedance analysis method [12–14]. Among them, the frequency sweep and the time domain simulation methods are used for verification analysis, whereas the eigenvalue and impedance analysis methods are used for system characteristic analysis and modeling. In the system characteristic analysis and modeling, the eigenvalue analysis method works based on the solution of the state equation. The increase in the size of the power system causes changes in the stable operating point or the system structure, requiring reconstruction of the equations of state. Moreover, the increase in the dimensionality of the linearized system state matrix leads to computational difficulties in solving the equations of state. In [10,11], the problems of SSO in the doubly-fed induction generator (DFIG) system are studied to calculate their eigenvalues and damping ratios, thus analyzing the dynamic characteristics of the system. Although the solutions are accurate, the processes of building the system state matrix using methods are complex and are accompanied by difficulties in calculating higher-order matrices. In contrast, studies using the impedance analysis method do not require the solution of higher-order equations of state or the addition or removal of unit modules. As only part of the equivalent impedance needs to be re-modeled, the impedance analysis method is most suitable for analyzing and modeling the mechanism of system SSO. In [12], an impedance modeling method based on the dq coordinate axis is proposed, revealing that the reason for the system instability is the negative impedance characteristic of the converter in the dq coordinate system, and the stability of the system is effectively improved by optimizing the phase-locked loop (PLL) parameters. The authors of [13] suggest that asymmetric modules such as the PLL and the outer loop of the controller make the system impedance model undergo frequency coupling. In [14], a positive–negative sequence impedance model of the DFIG considering frequency coupling characteristics is established, and the influences of each parameter on the stability of the system are analyzed. The literatures [12–14] have used the impedance model for parameters analysis and system stability optimization. However, these studies do not deliver effective SSO suppression strategies based on the established impedance models. Therefore, the risk of SSO of the system cannot be directly eliminated at the source.

At the present, a number of research results have been obtained on SSO suppression strategies. In general, these strategies can be divided into two categories, namely suppression using flexible ac transmission system (FACTS) devices [15,16] and suppression by introducing additional control in the converters [17–19]. In [15,16], the problems of SSO in DFIG systems are effectively solved by using FACTS devices. However, the introduction of these devices increased the hardware costs, resulting in poor cost–performance ratios. A novel adaptive several sub-synchronous control interaction (SSCI) mitigation method is proposed in [17], where the variable gain super-torsional sliding mode damping control parameters are optimized to eliminate SSO caused by the interaction of the rotor-side converter (RSC) and the series-compensated transmission line. However, the mechanism analysis of SSO is not sufficiently detailed and clear in this literature, and the key factors affecting oscillation are not clearly stated. In [18], a filtering and proportional differentiation-based damping controller is attached to RSC to add positive damping to the system, thus eliminating the SSO. However, analyses of the influences of relevant parameters such as the series compensation degree (SCD) and controller parameters on system stability are few. In [19], two damping controllers are attached to RSC and the grid-side converter (GSC), respectively, and the controllers' parameters are optimized under three different operating conditions, which suppresses the SSO and improves system stability. However, in the above strategy, SSO analysis and parameters adjustment are required in the case of changing operating conditions, resulting in the complex system analysis process and cumbersome parameters calculations.

This study proposes an SSO suppression strategy based on the attached virtual resistance controllers to RSC of a DFIG to further optimize the SSO suppression method. Several improvements are made in this study to address the limitations in the aforementioned studies. The highlights of the study are as follows:

- A DFIG equivalent impedance model considering RSC control is developed for the theoretical analysis of the system, which avoids complex higher-order state equation solutions and calculation of characteristic roots;
- The affecting factors and mechanisms of SSO in different phase sequences are analyzed in conjunction with the equivalent RLC resonant circuit of a DFIG-based series compensated grid-connected system (SCGCS), revealing that the causes of SSO in the system are the large equivalent negative resistance of RSC and rotor winding at the SSO frequency, resulting in a negative total system resistance;
- The influences of SCD and inner loop parameters (ILPs) on total system resistance-related SSO characteristics are analyzed, followed by parameters optimization to improve system stability. The analytical form of the total system resistance can be used directly for the ready optimization of SCD and ILPs;
- The attached virtual resistance controllers to RSC provide positive resistance to offset the negative resistance produced by RSC and rotor winding at the SSO frequency, thus eliminating oscillation. Additionally, no other additional devices are required, reducing hardware costs and attaining good economic benefits.

## 2. DFIG-Based Frequency Domain Impedance Model

The topology of a DFIG-based SCGCS in a doubly-fed wind farm is shown in Figure 1. This wind farm consists of multiple identical 5 MW DFIGs, each connected in parallel to the same bus by a 0.69/35 kV transformer T1. In addition, a stand-alone equivalent model is used to simulate the entire doubly-fed wind farm [1,13]. Moreover, this farm is connected to the 330 kV line by the 35/330 kV transformer T2, which is then connected in series with a compensation capacitor for long-distance transmission. Cg is the series compensation capacitor. The stator winding of the generator is connected to the grid via a transformer, whereas the rotor winding is ac-excited by means of the RSC and GSC regulation. The total output power of the generator is divided into two parts, i.e., the power fed to the grid on the stator side via the transformer and the power fed to the grid on the rotor side via the converter.

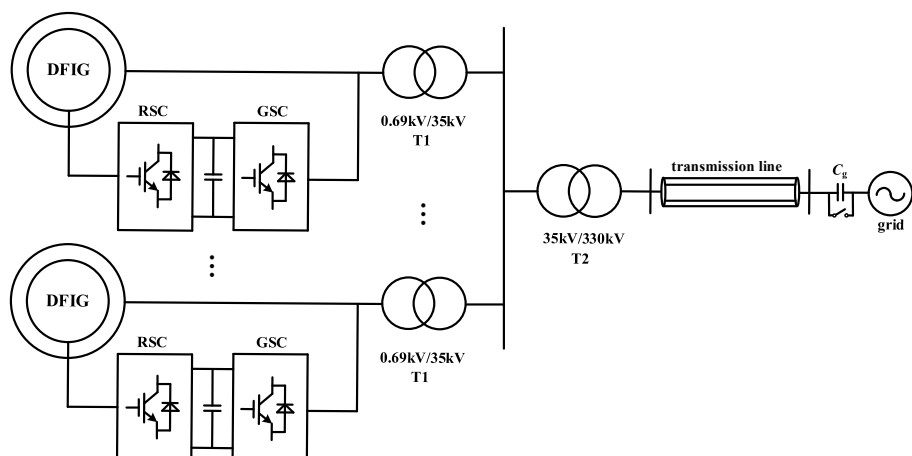

**Figure 1.** Topology of a SCGCS fin a doubly-fed wind farm.

In steady-state operation of DFIG, after converting the electrical variables and parameters from rotor side to the stator side, the T-shaped equivalent circuit of DFIG is shown in Figure 2 [20], and the relationships between stator-rotor voltage and current are shown in Equations (1)–(3):

$$\dot{U}_s = R_s \dot{I}_s + j\omega_1 L_s \dot{I}_s + j\omega_1 L_m \dot{I}_r \tag{1}$$

$$\varepsilon = \frac{\omega_1 - \omega_r}{\omega_1} \tag{2}$$

$$\frac{\dot{U}_r}{\varepsilon} = \frac{R_r}{\varepsilon}\dot{I}_r + j\omega_1 L_r \dot{I}_r + j\omega_1 L_m \dot{I}_s \tag{3}$$

where $\dot{U}_s$ and $\dot{I}_s$ are stator voltage and current, respectively; $\dot{U}_r$ and $\dot{I}_r$ are rotor voltage and current, respectively; $R_s$ and $R_r$ are stator resistance and rotor resistance, respectively; $\omega_1$ and $\omega_r$ are synchronous angular frequency and rotor angular frequency, respectively; $\varepsilon$ is the slip speed; $L_m$ is the mutual inductance between stator winding and rotor winding; $L_{1s}$ and $L_s$ are stator leakage inductance and self-inductance, respectively; $L_{1r}$ and $L_r$ are rotor leakage inductance and self-inductance, respectively. Additionally, $L_s = L_m + L_{1s}$, $L_r = L_m + L_{1r}$.

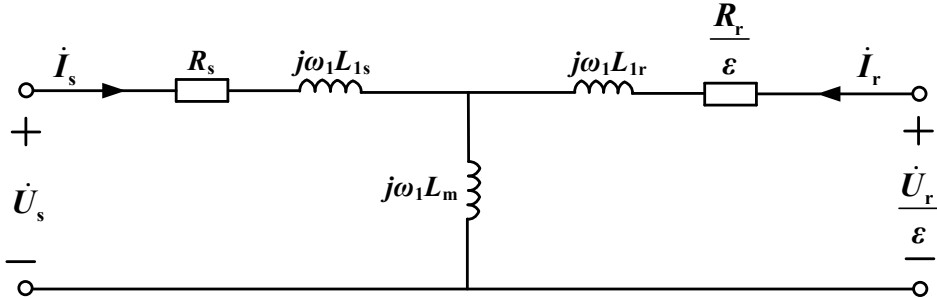

**Figure 2.** T-shaped equivalent circuit of DFIG.

### 2.1. Frequency Domain Expressions for Variables Considering Phase Sequence in the Dq Coordinate System

In the case where the stator voltage contains a small signal disturbance voltage component with the frequency of $\omega_e$, the frequency domain expressions of the A-phase voltage fundamental voltage component $\dot{U}_{sA1}$ and disturbance voltage component $\dot{U}_{sAe}$ in three-phase stationary coordinate are shown in Equation (4) [21]:

$$\begin{cases} \dot{U}_{sA1} = \dot{U}_1, \omega = \omega_1 \\ \dot{U}_{sAe} = \dot{U}_e, \omega = \omega_e \end{cases} \tag{4}$$

where $\dot{U}_1$ and $\dot{U}_e$ are the frequency domain expressions for the fundamental voltage component and disturbance voltage component, respectively, ( $\dot{U}_1 = U_1 e^{j\varphi 1}$, $\dot{U}_e = U_e e^{j\varphi 1}$). Similarly, the frequency domain expressions for the B-phase and C-phase voltages can also be derived.

The frequency domain expressions for the $\dot{U}_{sder}$ and $\dot{U}_{sqer}$ of the perturbation components in the dq coordinate system after the park transform are shown in Equation (5) [21]:

$$\begin{cases} \dot{U}_{sder} = \dot{U}_e \\ \dot{U}_{sqer} = (-1)^n j\dot{U}_e \end{cases}, \omega = \omega_e + (-1)^n \omega_1 \tag{5}$$

where $\dot{U}_e$ and $(-1)^n j\dot{U}_e$ are the frequency domain expressions for the stator disturbance voltage components in the d and q axes, respectively; $n$ is the phase sequence mark ($n = 1$ for positive sequence and $n = 0$ for negative sequence).

### 2.2. DFIG-Based Frequency Domain Impedance Model Considering Small Signal Perturbations in the Three-Phase Stationary Coordinate System

In the three-phase stationary coordinate system, the relationship between the A-phase stator disturbance voltage $\dot{U}_{sAe}$ and the stator disturbance current $\dot{I}_{sAe}$ and rotor disturbance current $\dot{I}_{rae}$ can be obtained from Equations (1)–(3), as shown in Equation (6).

Additionally, the relationship between the a-phase rotor disturbance voltage $\dot{U}_{\text{rae}}$ and the stator disturbance current $\dot{I}_{\text{sAe}}$ and rotor disturbance current $\dot{I}_{\text{rae}}$ is shown in Equation (7):

$$\dot{U}_{\text{sAe}} = (R_{\text{s}} + j\omega_{\text{e}}L_{\text{s}})\dot{I}_{\text{sAe}} + j\omega_{\text{e}}L_{\text{m}}\dot{I}_{\text{rae}} \tag{6}$$

$$\frac{\dot{U}_{\text{rae}}}{\varepsilon_{\text{er}}} = \left(\frac{R_{\text{r}}}{\varepsilon_{\text{er}}} + j\omega_{\text{e}}L_{\text{r}}\right)\dot{I}_{\text{rae}} + j\omega_{\text{e}}L_{\text{m}}\dot{I}_{\text{sAe}} \tag{7}$$

where $\omega_{\text{er}}$ is the slip at the $\omega_{\text{e}}$ frequency ($\omega_{\text{er}} = \omega_{\text{e}} + (-1)^n\omega_{\text{r}}$); $\varepsilon_{\text{er}}$ is the slip speed at the $\omega_{\text{e}}$ frequency ($\varepsilon_{\text{er}} = (\omega_{\text{e}} + (-1)^n\omega_{\text{r}})/\omega_{\text{e}}$).

To obtain the DFIG input impedance, the rotor is equated to a short circuit and the circuit topology is obtained with only the stator voltage acting. Let $\dot{U}_{\text{rae}} = 0$ to divide out the rotor current. The DFIG frequency domain input impedance $Z_{\text{dfig}}$ under small signal disturbance is obtained from Equations (6) and (7), as shown in Equation (8):

$$Z_{\text{dfig}} = \frac{\dot{U}_{\text{sAe}}}{\dot{I}_{\text{sAe}}} = R_{\text{s}} + j\omega_{\text{e}}L_{\text{s}} - \frac{(j\omega_{\text{e}}L_{\text{m}})^2}{(R_{\text{r}}/\varepsilon_{\text{er}}) + j\omega_{\text{e}}L_{\text{r}}} = R_{\text{s}} + R_{\text{r}}A_1\varepsilon_{\text{er}} + j\omega_{\text{e}}L_{\text{s}} - j\omega_{\text{er}}L_{\text{r}}A_1\varepsilon_{\text{er}} \tag{8}$$

where $A_1 = (\omega_{\text{e}}L_{\text{m}})^2/[R_{\text{r}}^2 + (\omega_{\text{er}}L_{\text{r}})^2]$.

### 2.3. DFIG-Based Frequency-Domain Impedance Model Considering Rsc in the Three-Phase Stationary Coordinate System

When DFIG operates stably, the dc bus voltage remains stable and the controls of RSC and GSC are decoupled. The control of the RSC achieves decoupling of the DFIG active power and reactive power, and its dynamic control process will affect the stability of the system. The large time constant and the slow dynamic adjustment speed of RSC's power outer loop lead to a smaller impact on SSO. Therefore, the derivation of the DFIG-based frequency-domain impedance model for RSC is carried out by considering only the role of the RSC's current inner loop [18,21]. RSC's current inner loop control-related block diagram in the dq coordinate system is shown in Figure 3, where the input and output are the rotor current reference and rotor voltage given in the dq axis, respectively. The relationship between the rotor voltage $u_{\text{rd}}$ and $u_{\text{rq}}$ and rotor current $i_{\text{rd}}$ and $i_{\text{rq}}$ components in the converter in the dq axis is shown in Equation (9):

$$\begin{cases} u_{\text{rd}} = \left(K_{\text{p1}} + \frac{K_{\text{i1}}}{s}\right)(i_{\text{rdref}} - i_{\text{rd}}) - \omega_{\text{slip}}L_{\text{r}}i_{\text{rq}} \\ u_{\text{rq}} = \left(K_{\text{p1}} + \frac{K_{\text{i1}}}{s}\right)(i_{\text{rqref}} - i_{\text{rq}}) + \omega_{\text{slip}}L_{\text{r}}i_{\text{rd}} \end{cases} \tag{9}$$

where $\omega_{\text{slip}}$ is the slip at the $\omega_1$ frequency ($\omega_{\text{slip}} = \omega_1 - \omega_{\text{r}}$); $K_{\text{p1}}$ and $K_{\text{i1}}$ are the ratio and integration coefficient of RSC's current inner loop, respectively; $i_{\text{rdref}}$ and $i_{\text{rqref}}$ are the current reference values in d and q axis, respectively.

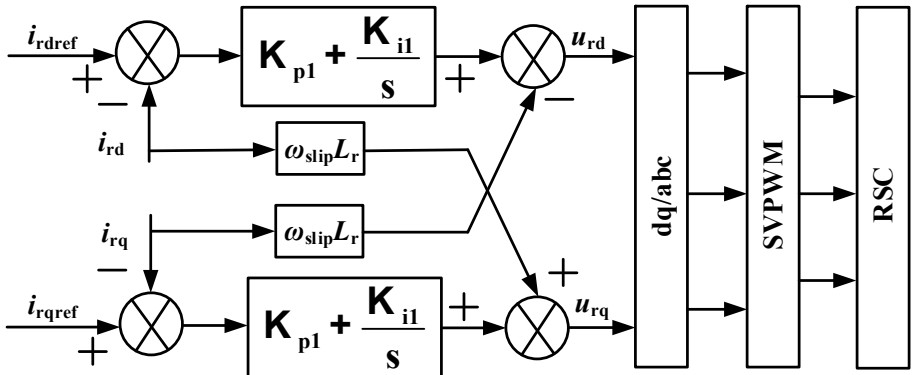

**Figure 3.** RSC's current inner loop control-related block diagram.

The relationship between the rotor disturbance voltage $\dot{U}_{\mathrm{rde}}$ and $\dot{U}_{\mathrm{rqe}}$ and the rotor disturbance current $\dot{I}_{\mathrm{rde}}$ and $\dot{I}_{\mathrm{rqe}}$ in the dq-axis in the frequency domain form via the RSC current inner loop, after accounting for small signal disturbances and performing harmonic linearization, is shown in Equation (10):

$$\begin{cases} \dot{U}_{\mathrm{rde}} = -\left(K_{\mathrm{p1}} - j\frac{K_{\mathrm{i1}}}{\omega_{\mathrm{er}}}\right)\dot{I}_{\mathrm{rde}} - \omega_{\mathrm{er}}L_{\mathrm{r}}\dot{I}_{\mathrm{rqe}} \\ \dot{U}_{\mathrm{rqe}} = -\left(K_{\mathrm{p1}} - j\frac{K_{\mathrm{i1}}}{\omega_{\mathrm{er}}}\right)\dot{I}_{\mathrm{rqe}} + \omega_{\mathrm{er}}L_{\mathrm{r}}\dot{I}_{\mathrm{rde}} \end{cases} \tag{10}$$

Equation (10) is transformed to a three-phase stationary coordinate system by a coordinate transformation, and the rotor side is converted to the stator side. At this point, the relationship between the a-phase RSC voltage $\dot{U}_{\mathrm{rae}}$ and the rotor current $\dot{I}_{\mathrm{rae}}$ in the stationary coordinate system is shown in Equation (11):

$$\begin{cases} \dfrac{\dot{U}_{\mathrm{rae}}}{\varepsilon_{\mathrm{er}}} = \dfrac{-\left(K_{\mathrm{p1}} - (jK_{\mathrm{i1}}/\omega_{\mathrm{er}}) - j\omega_{\mathrm{er}}L_{\mathrm{r}}\right)}{\varepsilon_{\mathrm{er}}}\dot{I}_{\mathrm{rae}} \\ Z_{\mathrm{RSC}} = \dfrac{\dot{U}_{\mathrm{rae}}}{\dot{I}_{\mathrm{rae}}} = -\left(K_{\mathrm{p1}} + (jK_{\mathrm{i1}}/\omega_{\mathrm{er}}) + j\omega_{\mathrm{er}}L_{\mathrm{r}}\right) \end{cases} \tag{11}$$

where $Z_{\mathrm{RSC}}$ is the equivalent impedance of RSC.

Combining Equations (6), (8) and (11) divides out rotor voltage and current. The DFIG input impedance $Z_{\mathrm{rdfig}}$ after considering the rotor controller is as follows:

$$Z_{\mathrm{rdfig}} = \frac{\dot{U}_{\mathrm{sAe}}}{\dot{I}_{\mathrm{sAe}}} = R_{\mathrm{s}} + j\omega_{\mathrm{e}}L_{\mathrm{s}} - \frac{(j\omega_{\mathrm{e}}L_{\mathrm{m}})^2}{\frac{R_{\mathrm{r}}}{\varepsilon_{\mathrm{er}}} + j\omega_{\mathrm{e}}L_{\mathrm{r}} + \frac{Z_{\mathrm{RSC}}}{\varepsilon_{\mathrm{er}}}} = R_{\mathrm{s}} + A_2(R_{\mathrm{r}} + K_{\mathrm{p1}})\varepsilon_{\mathrm{er}} + j\omega_{\mathrm{e}}L_{\mathrm{s}} - jA_2(K_{\mathrm{i1}}/\omega_{\mathrm{er}} + 2\omega_{\mathrm{er}}L_{\mathrm{r}})\varepsilon_{\mathrm{er}} \tag{12}$$

where $A_2 = (\omega_{\mathrm{e}}L_{\mathrm{m}})^2/[(R_{\mathrm{r}} + K_{\mathrm{p1}})^2 + ((K_{\mathrm{i1}}/\omega_{\mathrm{er}}) + 2\omega_{\mathrm{er}}L_{\mathrm{r}})^2]$.

The equivalent resistance $R_{\mathrm{rdfig}}$ of the DFIG after considering the rotor converter can be obtained from Equation (12), as shown in Equation (13):

$$R_{\mathrm{rdfig}} = Re\left(Z_{\mathrm{rdfig}}\right) = R_{\mathrm{s}} + A_2(R_{\mathrm{r}} + K_{\mathrm{p1}})\varepsilon_{\mathrm{er}} = R_{\mathrm{s}} + \frac{(\omega_{\mathrm{e}}L_{\mathrm{m}})^2(R_{\mathrm{r}} + K_{\mathrm{p1}})\varepsilon_{\mathrm{er}}}{\left(R_{\mathrm{r}} + K_{\mathrm{p1}}\right)^2 + ((K_{\mathrm{i1}}/\omega_{\mathrm{er}}) + 2\omega_{\mathrm{er}}L_{\mathrm{r}})^2} \tag{13}$$

## 3. Mechanism Analysis of Sub-Synchronous Oscillation Induced by Series Compensation and Grid Connection of DFIG

The DFIG-based SCGCS can be equated to an RLC series resonant circuit around the SSO frequency. Therefore, the SSO frequency of the system can be determined using the frequency-impedance curve of the input impedance. Firstly, the SSO frequency of the system is obtained by the frequency-resistance curve, with the frequency corresponding to the zero-crossing point (ZCP) being the SSO frequency. Subsequently, the system stability is determined from the frequency-resistance curve. If the equivalent resistance at the oscillation frequency is positive, the system will be stable due to the presence of positive damping; if the equivalent resistance at the oscillation frequency is negative, the system will oscillate and continue to diverge [22,23]. To analyze the SSO triggered by series compensation and grid connection, the RLC series resonant equivalent circuit is used to analyze the effects of different phase sequences, RSC parameters, and grid parameters on the SSO based on the established frequency domain impedance model. The DFIG parameters are shown in Table 1 and the transmission line and grid parameters are shown in Table 2. the distribution transmission line has a voltage level of 330 kV and a length of 100 km, where the equivalent resistors $R_{\mathrm{L}}$, inductance $L_{\mathrm{L}}$, conductivity $G_{\mathrm{u}}$ and distributed capacitance $G_{\mathrm{u}}$ of the unit length line are taken from the typical parameters of the transmission line at the voltage level of 330 kV [24].

**Table 1.** Parameters of DFIG.

| Parameters | Values and Units |
|---|---|
| Rated power ($P_n$) | 5 MW |
| system frequency ($f$) | 50 Hz |
| dc bus capacity ($C_{dc}$) | 50,000 [uF] |
| $U_s$ | 0.69 kV |
| $R_r$ | 0.00607 [pu] |
| $R_s$ | 0.0054 [pu] |
| $L_{lr}$ | 0.11 [pu] |
| $L_{ls}$ | 0.1 [pu] |
| $L_m$ | 4.5 [pu] |
| $\omega_r$ | 1 [pu] |

**Table 2.** Parameters of DFIG Transmission line and grid parameters.

| Parameters. | Values and Units |
|---|---|
| $R_Z$ | 0.046346 [pu] |
| $L_Z$ | 0.371263 [pu] |
| $G_u$ | 3.821374 [pu] |
| $C_u$ | 0.002695 [pu] |
| $C_g$ | 0.002681 [pu] |
| $R_g$ | 0.010214 [pu] |
| $L_g$ | 0.085473 [pu] |
| $l$ | 100 km |

*3.1. Analysis of Sub-Synchronous Oscillation Based on the RLC Series Resonant Equivalent Circuit*

Figure 4 shows the equivalent model of transmission line distribution parameters where $l$ is the length of the transmission line, and $R_L$, $L_L$, $G_u$, and $C_u$ are equivalent resistors, inductors, conductivities and distributed capacitors per unit length line, respectively. $V_{in}$ and $I_{in}$ are the input voltage and current, respectively.

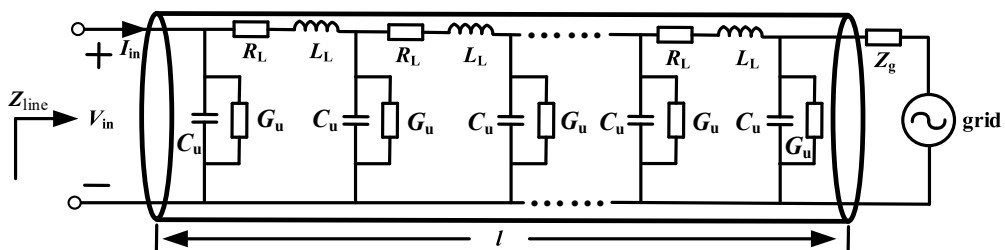

**Figure 4.** Distributed parameter equivalent model of transmission line.

The equivalent input impedance $Z_{line}$ seen from the head end of the line in Figure 4 is shown in Equation (14). $Z_{line}$ represents the equivalent impedance of transmission line and power grid:

$$Z_{line} = \frac{V_{in}}{I_{in}} = \frac{Z_g + Z_c \tanh(\gamma l)}{1 + \frac{Z_g}{Z_c} \tanh(\gamma l)} \tag{14}$$

where $Z_g$ is the network side impedance, $Z_c$ and $\gamma$ are line wave impedance and propagation coefficient, and the expressions for $Z_g$, $Z_c$, and $\gamma$ are shown in Equation (15):

$$\begin{cases} Z_g = R_g + j\omega L_g \\ Z_c = \sqrt{z/y} = \sqrt{(R_L + j\omega L_L)/(G_u + j\omega C_u)} \\ \gamma = \sqrt{zy} = \sqrt{(R_L + j\omega L_L)(G_u + j\omega C_u)} \end{cases} \tag{15}$$

where $R_g$ and $L_g$ are grid equivalent resistors and inductors, respectively; $z$ and $y$ are the unit length impedance and admittance of the line, respectively; $\omega$ is the angular frequency of the transmission signal.

The RLC series resonant equivalent circuit of the DFIG-based SCGCS is shown in Figure 5, Rrdfig and Lrdfig are the equivalent resistance and inductance of doubly-fed induction motor, respectively. $Z_{line}$ is the equivalent impedance of the transmission line and the grid, $C_g$ denotes the series compensation capacitance on the transmission line. $U_{grid}$ and $U_{dfig}$ denote the equivalent voltage source voltages of the grid and DFIG, respectively.

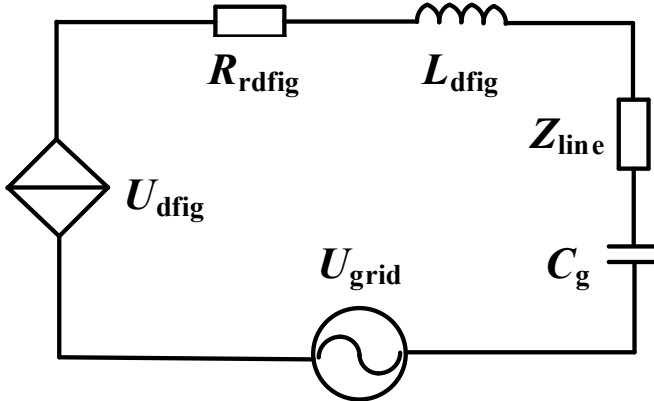

**Figure 5.** RLC series resonant equivalent circuit of the DFIG-based SCGCS.

In order to clearly analyze the influence of the relevant parameters on the subsynchronous oscillation of the system, the RLC series resonance equivalent circuit of the DFIG series complement grid-connected system needs to be further optimized. According to Equation (14), the equivalent impedance of transmission line and power grid is $Z_{line}$, and Equation (16) can be derived from $Z_{line}$:

$$Z_{line} = \frac{V_{in}}{I_{in}} = \frac{Z_g + Z_c \tanh(\gamma l)}{1 + \frac{Z_g}{Z_c}\tanh(\gamma l)} = \frac{D_1 B_1 + \omega^2 D_2 B_2}{B_1^2 + \omega^2 B_2^2} + \frac{j\omega(D_2 B_1 - D_1 B_2)}{B_1^2 + \omega^2 B_2^2} = R_Z + j\omega L_Z \tag{16}$$

where $R_Z = \frac{D_1 B_1 + \omega^2 D_2 B_2}{B_1^2 + \omega^2 B_2^2}$, $L_Z = \frac{D_2 B_1 - D_1 B_2}{B_1^2 + \omega^2 B_2^2}$, $B_1 = aR_g G_u + \gamma - a\omega^2 C_u L_g$, $B_2 = aR_g C_u + aL_g G_u$, $D_1 = \gamma R_g + aR_L$, $D_2 = \gamma R_g + aL_L$, $a = \tanh(\gamma l)$.

Based on this equivalent circuit, the equivalent system total inductance $L_{sum}$, total resistance $R_{sum}$, total reactance $X_{sum}$, SCD $k$, SSO frequency $f_{SSR}$, and SSO disturbance angular frequency $\omega_e$ of a DFIG connected to the grid through a series compensated transmission line are shown in Equations (17)–(22), respectively:

$$R_{sum} = R_{rdfig} + R_Z = Re\left(Z_{rdfig}\right) + R_Z \tag{17}$$

$$L_{sum} = L_{dfig} + L_Z = Im\left(Z_{rdfig}\right) + L_Z \tag{18}$$

$$X_{sum} = X_L + X_C = \omega L_{sum} - \frac{1}{\omega C_g} \tag{19}$$

$$k = \frac{X_C}{X_L} = \frac{1}{\omega^2 C_g L_{sum}} \tag{20}$$

$$f_{SSR} = \frac{1}{2\pi\sqrt{L_{sum}C_g}} = \frac{1}{2\pi\sqrt{\left(L_{dfig} + L_Z\right) \times C_g}} \tag{21}$$

$$\omega_{\mathrm{e}} = 2\pi f_{\mathrm{SSR}} = \frac{1}{\sqrt{L_{\mathrm{sum}} C_{\mathrm{g}}}} = \frac{1}{2\pi \sqrt{\left( L_{\mathrm{dfig}} + L_Z \right) C_{\mathrm{g}}}} \tag{22}$$

where $R_{\mathrm{rdfig}}$, $R_Z$ and $R_{\mathrm{g}}$ are the equivalent resistance for the DFIG, the transmission line and the grid, respectively; $L_{\mathrm{dfig}}$, $L_Z$ and $L_{\mathrm{g}}$ are the equivalent inductance for the DFIG, the transmission line and the grid, respectively; *Re* and *Im* are the real and imaginary components of the extracted $Z_{\mathrm{rdfig,}}$ respectively; $X_L$ and $X_C$ are the inductive and capacitive reactance of the transmission line, respectively.

The RLC series resonance equivalent circuit of the new DFIG series compensation grid connected system is shown in Figure 6. When the series compensation $k = 15\%$, $K_{\mathrm{p1}} = 1.5$, $K_{\mathrm{i1}} = 0.05$, the original manuscript Equations (17)–(19) and (22) can be combined to obtain $R_{\mathrm{sum}} = 0.030157$ [pu], $L_{\mathrm{sum}} = 4.734105$ [pu], $C_{\mathrm{g}} = 0.002681$ [pu].

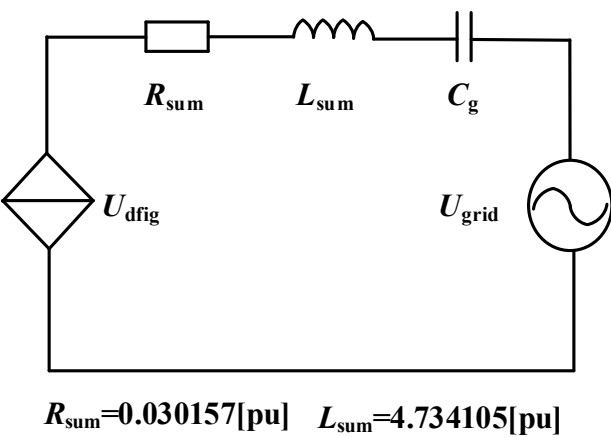

**$R_{\mathrm{sum}}$=0.030157[pu]**   **$L_{\mathrm{sum}}$=4.734105[pu]**

**$C_{\mathrm{g}}$=0.026801[pu]**

**Figure 6.** The new RLC series resonant equivalent circuit of the DFIG-based SCGCS.

According to Equations (13) and (17), at the SSO frequency, the equivalent resistance $R_{\mathrm{rdfig}}$ of a DFIG may be negative, with a value greater than the sum of the line resistance $R_Z$ and the grid equivalent resistance $R_{\mathrm{g}}$, resulting in $R_{\mathrm{sum}} < 0$. At this point, the whole system exhibits negative resistance characteristics, triggering SSO in the system.

*3.2. Analysis of Sub-Synchronous Oscillation in Different Phase Sequences*

The slip speed $\varepsilon_{\mathrm{er}}$ in positive phase sequence is shown in Equation (23):

$$\varepsilon_{\mathrm{er}} = \frac{\omega_{\mathrm{e}} - \omega_{\mathrm{r}}}{\omega_{\mathrm{e}}} \tag{23}$$

When $\varepsilon_{\mathrm{er}} < 0$, Equations (13) and (17) suggest $R_{\mathrm{rdfig}} < 0$, which may cause the equivalent total resistance $R_{\mathrm{sum}} < 0$. At this point, the whole system exhibits a negative resistance characteristic, leading to SSO.

The slip speed $\varepsilon_{\mathrm{er}}$ in negative phase sequence is shown in Equation (24):

$$\varepsilon_{\mathrm{er}} = \frac{\omega_{\mathrm{e}} + \omega_{\mathrm{r}}}{\omega_{\mathrm{e}}} \tag{24}$$

Equation (24) suggests that the value of $\varepsilon_{\mathrm{er}}$ is always greater than 0. Therefore, the equivalent resistance of the rotor winding and RSC in negative sequence is always positive at the SSO frequency. Additionally, Equations (13) and (17) indicate that the value of total equivalent resistance $R_{\mathrm{sum}}$ is also always greater than 0, and SSO will not occur. Therefore, the system is only studied in positive phase sequence.

### 3.3. Effect of System Parameters on Sub-Synchronous Oscillation in Positive Phase Sequence

The analytical expression of the total resistance and reactance of the system can be directly used to analyze the influence of system parameters on subsynchronous oscillation under positive phase sequence. When the total resistance of the system $R_{sum} = 0$, the system is in a critical stable state. Using the least square method to search the regulator can complete the selection of the best parameters [25,26]. This manuscript is based on the recursive augmented least square estimation algorithm to select the best parameters.

In [3], the values of line series compensation are $C = 10\%$, $C = 15\%$, $C = 20\%$, respectively. The effect of series compensation on system stability is analyzed and studied. Therefore, refer to [3], and the values of string complement are $k = 10\%$, $k = 15\%$, and $k = 18\%$, respectively. The corresponding string complement capacitor values are $C_g = 0.004021$ [pu] (11.3804 uF), $C_g = 0.002681$ [pu] (7.58694 uF), $C_g = 0.002236$ [pu] (6.32645 uF).

When the line series compensation $k = 15\%$, combined with (18), (19) and (22), based on the recursive augmented least square estimation method, the zero crossing point of the total reactance of the system can be calculated when $f = 34.1$ Hz, and $K_{p1} = 1.5132$, $K_{i1} = 0.0537$ can be estimated. The estimation results are shown in Figure 7.

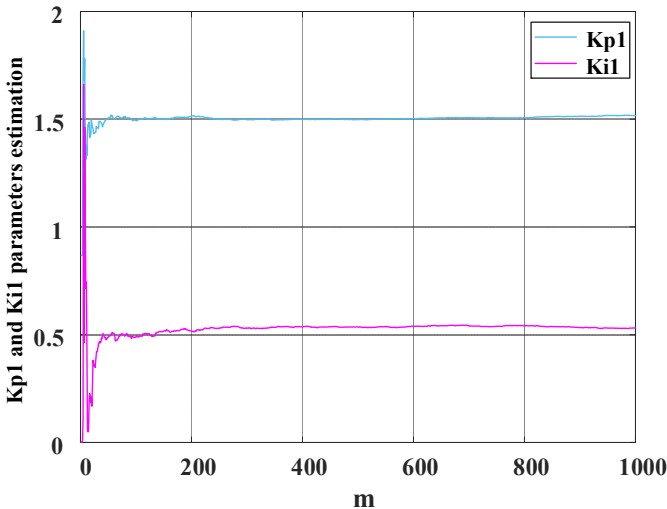

**Figure 7.** $K_{p1}$ and $K_{i1}$ parameters estimation.

Based on the system parameters of $K_{p1} = 1.5$, $K_{i1} = 0.05$, and $k = 15\%$, parameters adjustment near the critical stable state of the system can be performed to reveal the mechanism underlying the effect of parameter changes on the system. Therefore, changing SCD and ILPs can optimize the parameters to improve the system stability.

#### 3.3.1. Effect of Different SCDs on Sub-Synchronous Oscillation

When $K_{p1} = 1.5$ and $K_{i1} = 0.05$, $k$ is taken as 10%, 15%, and 18%, respectively. The frequency-impedance curves of the system derived from Equations (17)–(19) and (22) for three different SCDs are shown in Figure 8. At 10% SCD, the total system reactance undergoes a ZCP at 31.7 Hz. The total system resistance $R_{sum}$ at this frequency is greater than zero, so the system remains stable; at 15% SCD, the total system reactance undergoes a ZCP at 33.5 Hz and the total system resistance $R_{sum}$ at this frequency is greater than zero, so the system remains stable; at 18% SCD, the total system reactance undergoes a ZCP at 34.5 Hz. At this frequency, the total system resistance $R_{sum}$ is less than zero, causing the system to undergo SSO. Therefore, a larger SCD indicates a greater susceptibility to SSO.

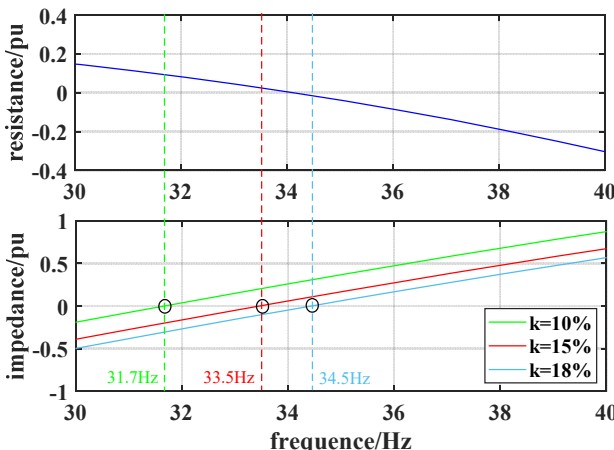

**Figure 8.** Frequency-impedance curves of the system at different SCDs.

### 3.3.2. Effect of Different Proportionality Coefficients on Sub-Synchronous Oscillation

When $K$ = 15% and $K_{i1}$ = 0.05, $k$ is taken as 0.8, 1.5, and 2, respectively. The system frequency-impedance curves obtained from Equations (17)–(19) and (22) for the three proportionality coefficients are shown in Figure 9. When $K_{p1}$ = 0.8, the total system reactance undergoes a ZCP at 32.3 Hz. The total system resistance $R_{sum}$ at this frequency is greater than zero, so the system remains stable; when $K_{p1}$ = 1.5, the total system reactance undergoes a ZCP at 33.5 Hz. The total system resistance $R_{sum}$ at this frequency is greater than zero, so the system remains stable; when $K_{p1}$ = 2, the total system reactance undergoes a ZCP at 35.2 Hz. The total system resistance $R_{sum}$ at this frequency is less than zero, causing the system to undergo SSO. Thus, a larger proportionality coefficient indicates a greater susceptibility to SSO.

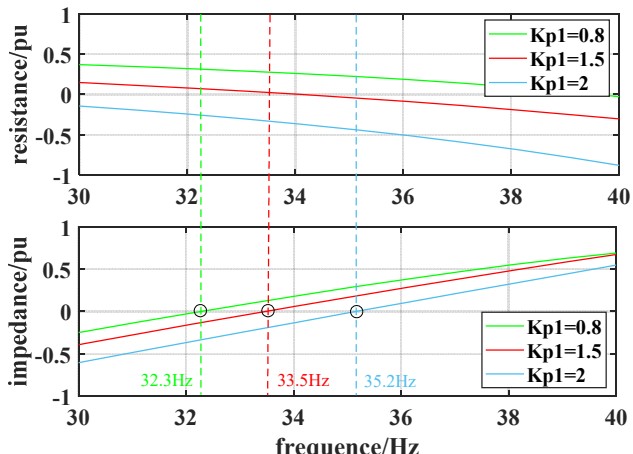

**Figure 9.** Frequency-impedance curves of the system with different proportionality coefficients.

### 3.3.3. Effect of Different Integration Coefficients on Sub-Synchronous Oscillation

When $K$ = 15% and $K_{p1}$ = 1.5, $K_{i1}$ is taken as 0.007, 0.05, and 0.1, respectively. The system frequency-impedance curves obtained from Equations (17)–(19) and (22) for three different integration coefficients are shown in Figure 10. When $K_{i1}$ = 0.007, the total system reactance undergoes a ZCP at 34.4 Hz. The total system resistance $R_{sum}$ at this frequency is less than zero, causing the system to undergo SSO. When $K_{i1}$ = 0.05, the total system reactance undergoes a ZCP at 33.5 Hz. The total system resistance $R_{sum}$ at this frequency is greater than zero, so the system remains stable; when $K_{i1}$ = 0.1, the total system reactance undergoes a ZCP at 33.2 Hz. The total system resistance $R_{sum}$ at this frequency is greater

than zero, so the system remains stable. Therefore, a smaller integration coefficient indicates a greater susceptibility to SSO.

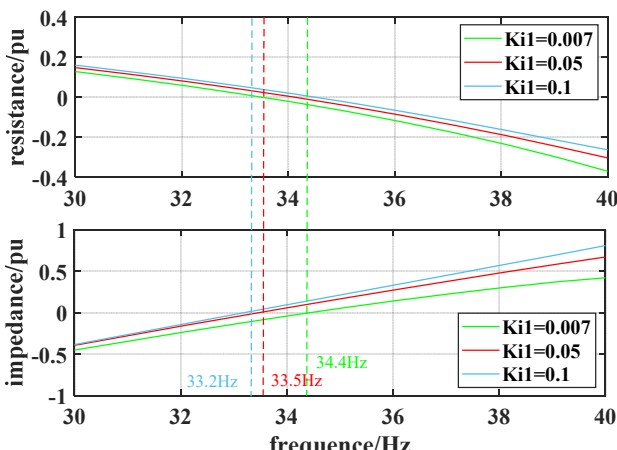

**Figure 10.** Frequency-impedance curves of the system with different integration coefficients.

The above analysis results suggest that SSO can be suppressed by parameters optimization. Additionally, SCDs of line should be minimized without reducing their transmission capacity. The proportionality coefficient $K_{p1}$ can be reduced and the integration coefficient $K_{i1}$ increased to enhance system stability. Nevertheless, an optimized suppression strategy is required to further eliminate the SSO at source.

## 4. Suppression Strategy of Attaching Virtual Resistance Controllers to RSC

Based on the mechanism of SSO in DFIG systems triggered by series compensation and grid connection, a SSO suppression strategy of attaching virtual resistance to RSC is proposed to eliminate the oscillation at source.

### 4.1. Structure of the Virtual Resistance Controllers

At the SSO frequency, a negative equivalent resistance $R_{rdfig}$ of RSC may cause the total system resistance $R_{sum}$ to be less than 0. At this point, the whole system has a negative resistance characteristic that triggers SSO. Therefore, at the SSO frequency, a positive resistance equal to the negative equivalent resistance $R_{rdfig}$ can be introduced as an attached virtual resistance to RSC to provide positive damping to the system, thus suppressing SSO.

The structure of the attached virtual resistance controllers to RSC is shown in Figure 11. The rotor currents $i_{rd}$ and $i_{rq}$ are introduced into the current inner loop of the RSC as inputs to the two controllers of the virtual resistance via the band-pass filter $G_{BP}$ and the proportional component $K_b$, respectively. The positive virtual voltage $\Delta u_r$ generated by the introduced $i_r$ offsets the negative disturbance voltage caused by the RSC control parameters at the SSO frequency.

### 4.2. Parameters Configuration of the Virtual Resistance Controllers

The proportional component $K_b$ makes the input $i_r$ linear with the virtual voltage $\Delta u_r$, generating a virtual resistance $R_v$. This virtual resistance can be used to compensate for the equivalent negative resistance generated by the rotor converter and the rotor winding, thus suppressing SSO in the system. Additionally, the equivalent resistance $R_{rdfig}$ can be offset directly by the resistance introduced by the virtual resistance controllers. Based on Equation (13), the value of the virtual resistance introduced is shown in Equation (25):

$$R_V = K_b = \Delta u_r / i_r = -\left[R_s + A_2 \left(R_r + K_{p1}\right) \varepsilon_{er}\right] \tag{25}$$

To reduce the influence of the virtual resistance controllers on the control characteristics of RSC, a band-pass filter is used to separate out the SSO components of the rotor current $i_r$. The filter is a second-order band-pass filter with the transfer function shown in Equation (26):

$$G_{BP}(s) = \frac{s/\omega_0}{(s/\omega_0)^2 + (2\xi s/\omega_0) + 1} \tag{26}$$

where $\omega_0$ is the center frequency of the filter (corresponding to the SSO frequency); $\xi$ is the damping coefficient ($\xi = 2$). $\xi$ is taken with reference to the literature [18,27].

Based on the analysis results of the effect of system parameters on SSO in positive phase sequence in Section 3.3 the SSO frequency and the equivalent resistance $R_{rdfig}$ when SSO occurs can be obtained. The center frequency $\omega_0$ of the filter and the virtual resistance $R_v$ are dynamically adjusted to offset the equivalent resistance $R_{rdfig}$, thus suppressing SSO.

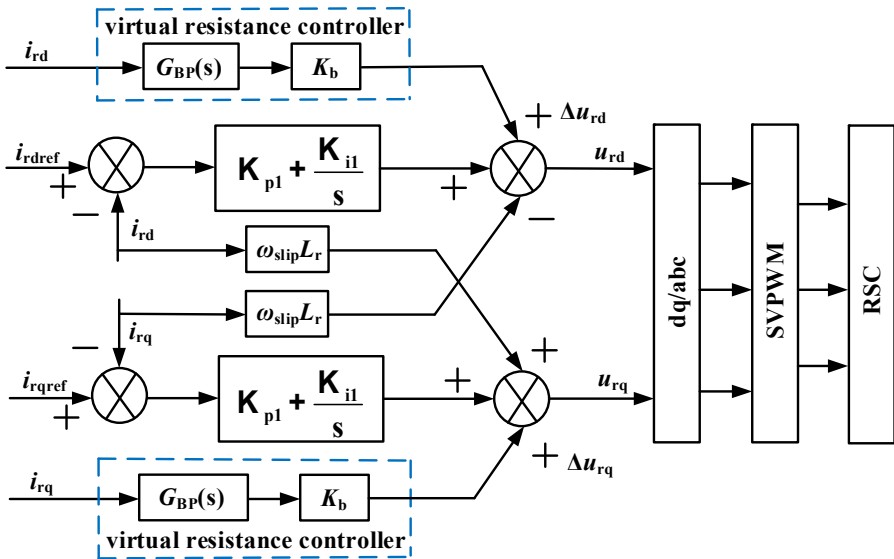

**Figure 11.** Structure of the virtual resistance controllers on the RSC.

## 5. Simulation Analysis

In this study, the time domain simulation method is used to study the influences of system parameters on system stability. In the analysis of system subsynchronous oscillation, FFT is usually used for transient waveform analysis [15,16,27]. The analysis results of the active power waveform and its fast Fourier transform (FFT) output from DFIG are obtained by introducing a series compensation capacitor at 3.5 s. It is verified that the system stability can be improved by optimizing the parameters. When SSO occur in the system at 3.5 s, the analysis results of the active power waveform and its FFT with and without the virtual resistance controllers are compared to verify the effectiveness of the proposed suppression strategy.

### 5.1. Effect of System Parameters on Sub-Synchronous Oscillation

5.1.1. Effect of Different SCDs on Sub-Synchronous Oscillation

When $K_{p1} = 1.5$ and $K_{i1} = 0.05$, $k$ is taken as 10%, 15%, and 18%, respectively. The analysis results of active power waveform and its FFT obtained by system simulation with different SCDs are shown in Figure 12. The system oscillation converges and finally stabilizes when the SCD is 10% and 15%, respectively; when the SCD is 18%, the system oscillation diverges. Figure 12b indicates that an 18% SCD leads to an SSO frequency component at 34.5 Hz. Thus, a larger SCD indicates a greater susceptibility to SSO. Therefore, the results obtained in Section 3.3.1 are verified.

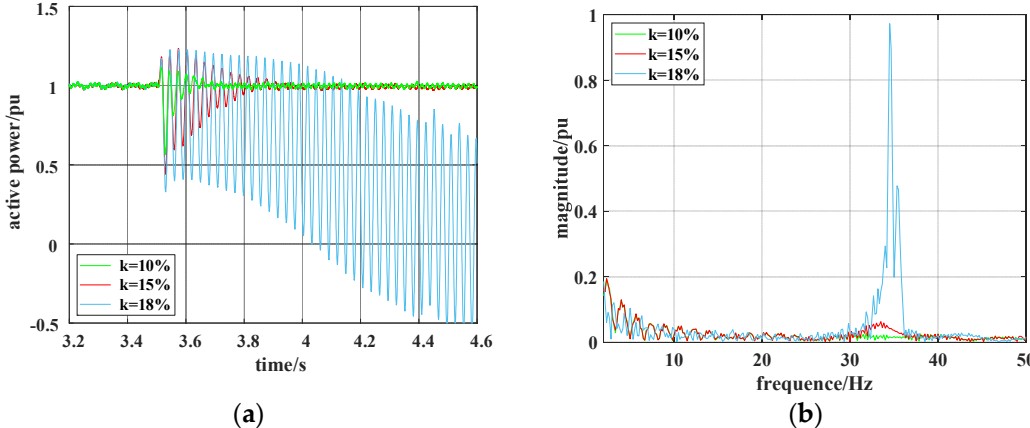

**Figure 12.** Effect of different SCDs on sub-synchronous oscillation: (**a**) Comparison of active power output from DFIG with different SCDs; (**b**) Comparison of FFT frequency of active power output from DFIG with different SCDs.

5.1.2. Effect of Different Proportionality Coefficients on Sub-Synchronous Oscillation

When $K_{i1} = 0.05$ and $k = 15\%$, $K_{p1}$ is taken as 0.8, 1.5, and 2, respectively. The analysis results of the active power waveform and its FFT obtained by system simulation with different proportionality coefficients are shown in Figure 13. When $K_{p1}$ is 0.8 and 1.5, respectively, the system oscillation converges and the system is stable at the end; when $K_{p1} = 2$, the system oscillation diverges. Figure 13b indicates a SSO frequency component at 35.2 Hz when $K_{p1} = 2$. Thus, a larger proportionality coefficient indicates a greater susceptibility to SSO. Therefore, the results obtained in Section 3.3.2 are verified.

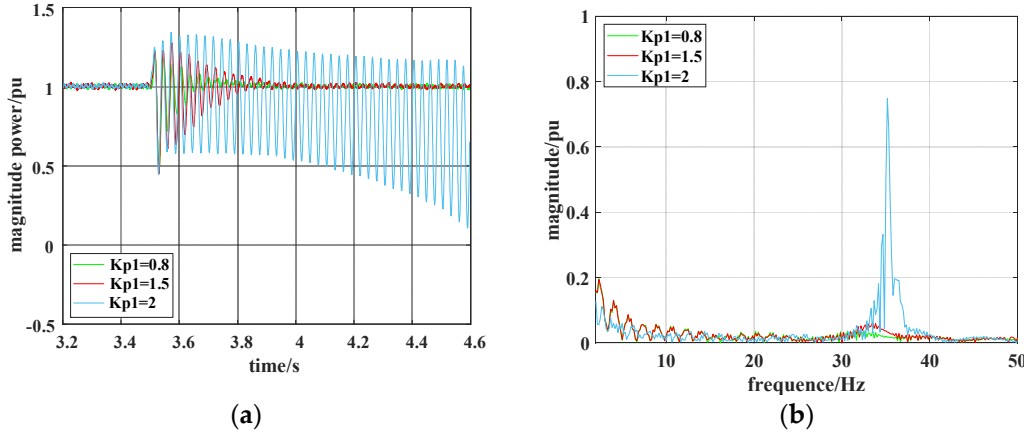

**Figure 13.** Effect of different proportionality coefficients on sub-synchronous oscillation: (**a**) Comparison of active power output from DFIG with different proportionality coefficients; (**b**) Comparison of FFT frequency of active power output from DFIG with different proportionality coefficients.

5.1.3. Effect of Different Integration Coefficients on Sub-Synchronous Oscillation

When $K_{p1} = 1.5$ and $k = 15\%$, $K_{i1}$ is taken as 0.007, 0.05, and 0.1, respectively. The analysis results of the active power waveform and its FFT obtained by system simulation with different integration coefficients are shown in Figure 14. When $K_{i1}$ is 0.007, the system oscillation diverges; when $K_{i1}$ is 0.8 and 1.5, the system oscillation converges and the system is stable at the end. Figure 14b indicates an SSO frequency component at 34.4 Hz when $K_{i1} = 0.007$. Thus, a lesser integration coefficient indicates a greater susceptibility to SSO. Therefore, the results obtained in Section 3.3.3 are verified.

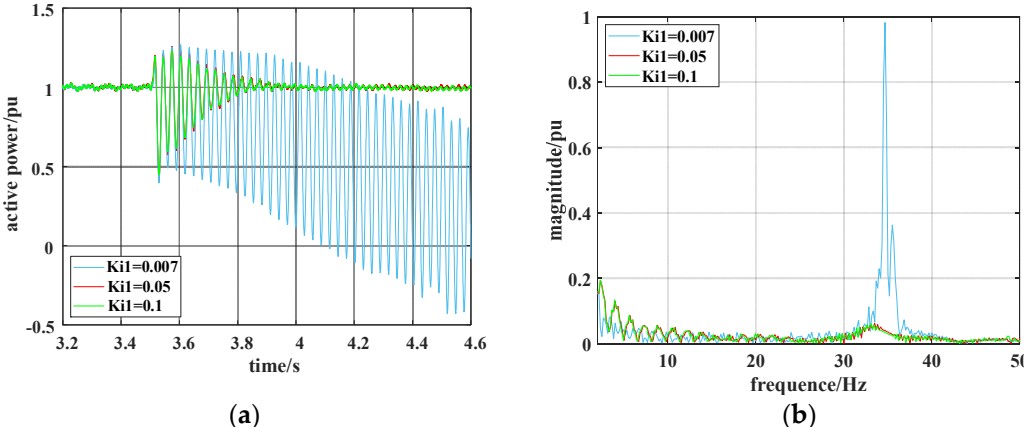

**Figure 14.** Effect of different integration coefficients on sub-synchronous oscillation: (**a**) Comparison of active power output from DFIG with different integration coefficients; (**b**) Comparison of FFT frequency of active power output from DFIG with different integration coefficients.

### 5.2. Suppression Results of Virtual Resistance Controllers

Section 3.3 shows that SSO occurs when SCDs, proportionality coefficients, and integration coefficients meet the following three cases, respectively: Case 1: $k$ = 18%, $K_{p1}$ = 1.5, and $K_{i1}$ = 0.05; Case 2: $k$ = 15%, $K_{p1}$ = 2, and $K_{i1}$ = 0.05; Case 3: $k$ = 15%, $K_{p1}$ = 1.5, and $K_{i1}$ = 0.007. The virtual resistance controllers attached suppress the SSO occurring in the system in these three cases. The analysis results of the active power waveform and its FFT frequency with and without the virtual resistance controllers for the three oscillation cases are shown in Figure 15. Figure 15a, c and e suggest that the active power oscillation diverges when no suppression is applied. After the introduction of the virtual resistance controllers, the active power output from the system gradually converges and remains stable. Additionally, Figure 15 suggests that the SSO frequencies for the three oscillation cases are 34.5, 35.2, and 34.4 Hz, respectively, when no suppression is applied. Then, no SSO occurs in the system after the introduction of the virtual resistance controllers. Therefore, these results verify the effectiveness of the suppression strategy.

### 5.3. Considering the Effect of Magnetic Saturation on Subsynchronous Oscillation

Under different $k$, different $K_{p1}$ and different $K_{i1}$, the effect of core magnetic saturation of DFIG on system stability is analyzed by simulation. Figure 16 compares the active power of the system with and without magnetic saturation under different $k$, Figure 17 compares the active power of the system with and without magnetic saturation under different $K_{p1}$, and Figure 18 compares the active power of the system with and without magnetic saturation under different $K_{i1}$.

As can be seen from Figures 16–18, under the same control strategy, the simulation results of the linear mathematical model considering magnetic circuit saturation are compared and analyzed, and the following conclusions are drawn: (1) The influence of magnetic saturation on the operation performance of DFIG under normal operation state is very small, which can be ignored; (2) When subsynchronous oscillation occurs, magnetic saturation has a great impact on the operation performance of DFIG, and the system will oscillate greatly in a short time, which is larger than the power fluctuation of linear mathematical model. Using effective suppression strategy can still suppress the oscillation of the system; (3) In the actual operation of DFIG, the magnetic circuit saturation exists, and it is of practical significance to consider the magnetic saturation in the system transient analysis.

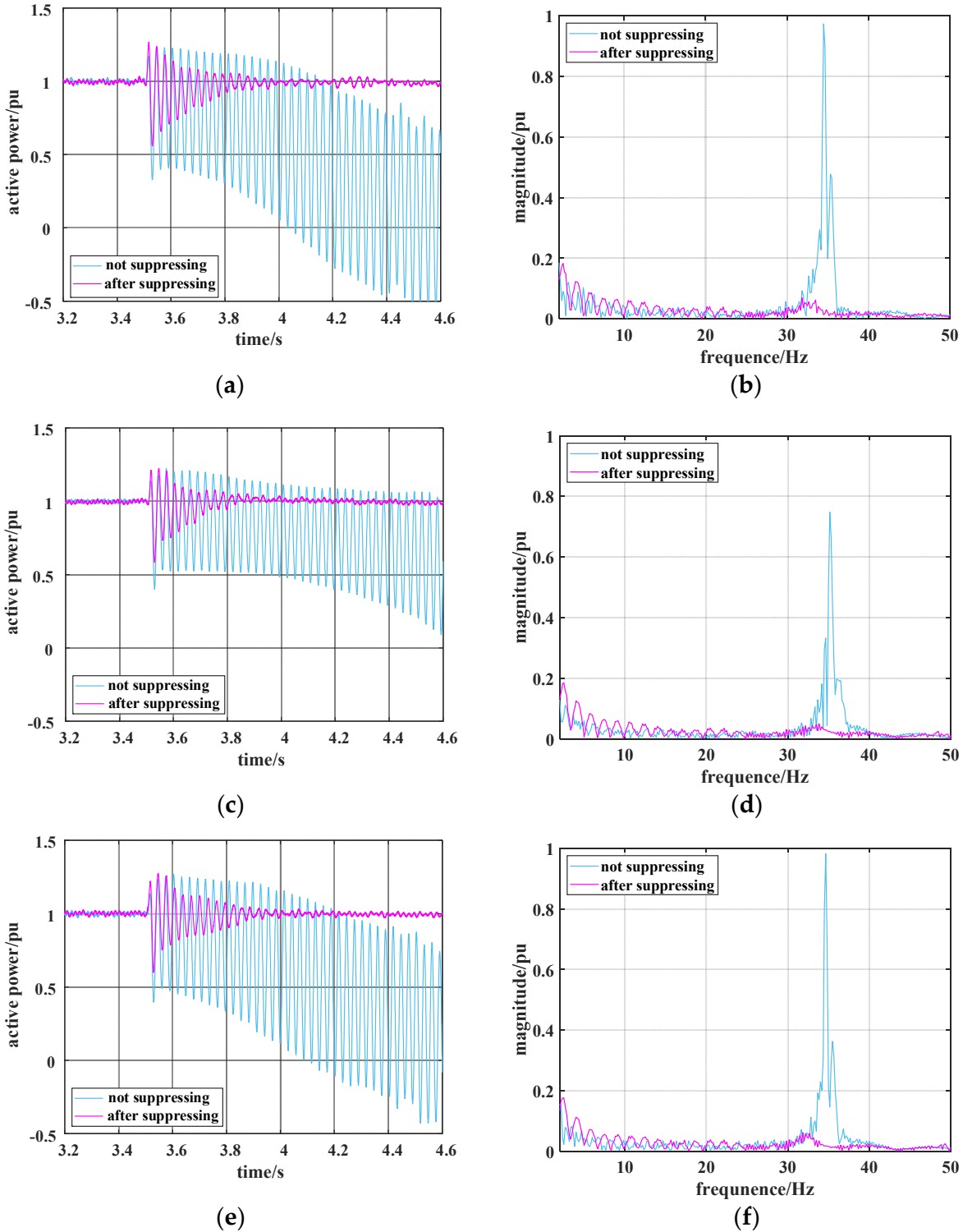

**Figure 15.** Suppression results of virtual resistance controllers: (**a**) Case 1: Comparison of active power output from DFIG before and after suppression; (**b**) Case 1: Comparison of FFT before and after suppression; (**c**) Case 2: Comparison of active power output from DFIG before and after suppression; (**d**) Case 2: Comparison of FFT before and after suppression; (**e**) Case 3: Comparison of active power output from DFIG before and after suppression; (**f**) Case 3: Comparison of FFT before and after suppression.

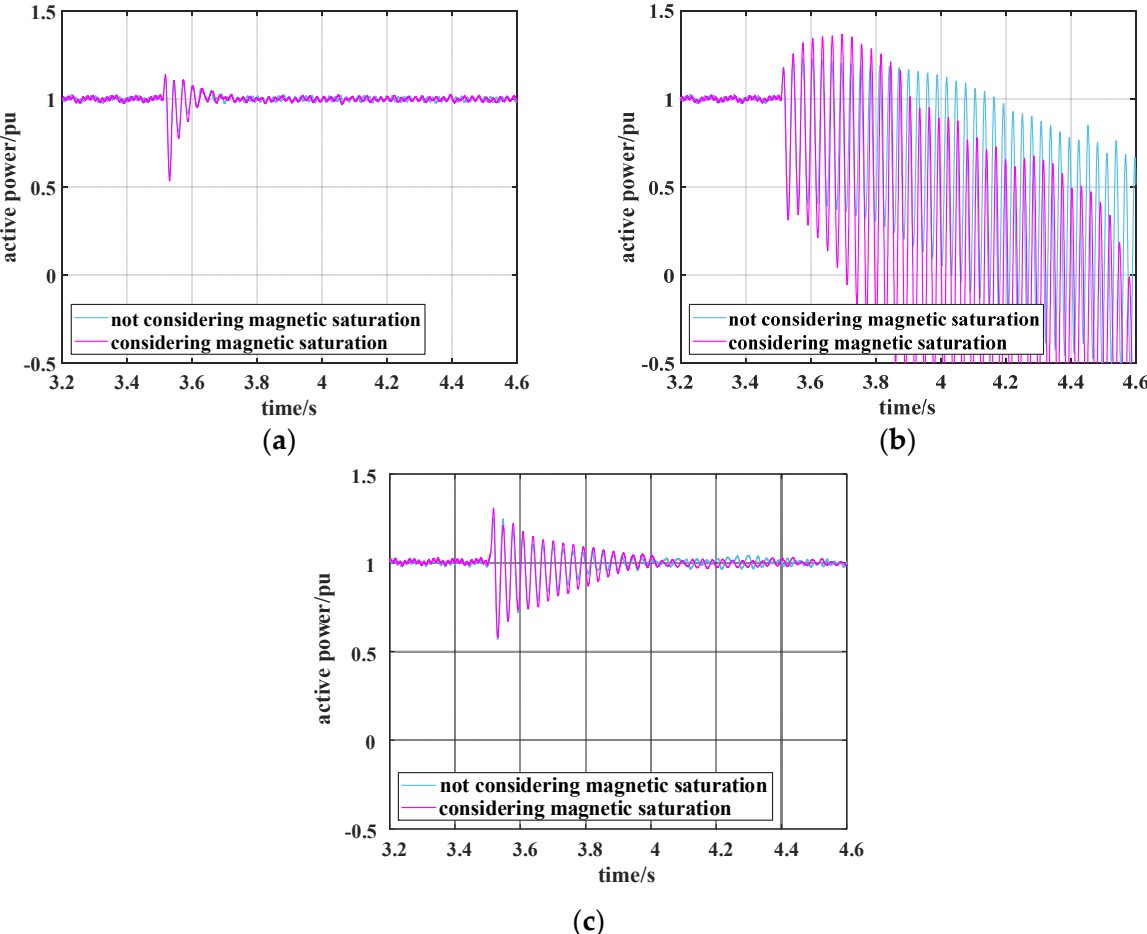

**Figure 16.** Comparison of the active power of the system with and without magnetic saturation under different *k*: (**a**) when *k* = 10%, comparison of the active power of the system with and without magnetic saturation; (**b**) when *k* = 18%, comparison of the active power of the system with and without magnetic saturation; (**c**) when *k* = 18%, comparison of subsynchronous suppression of the system with and without considering magnetic saturation.

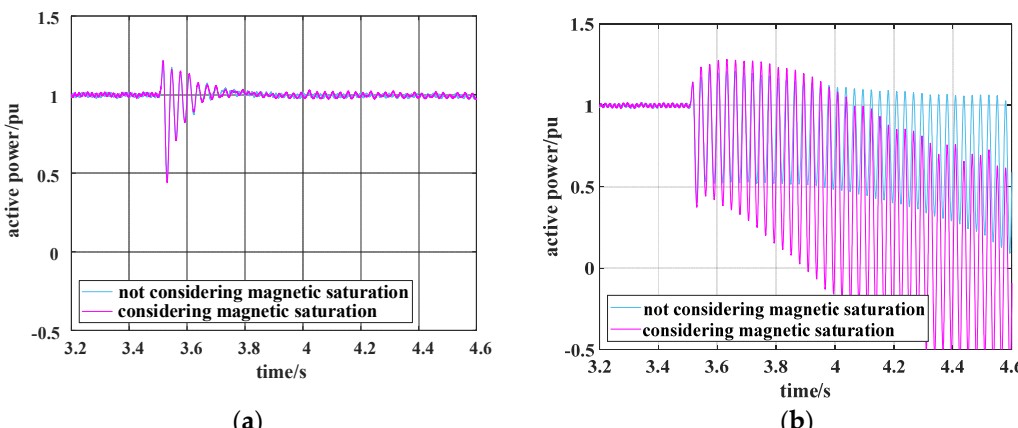

**Figure 17.** *Cont.*

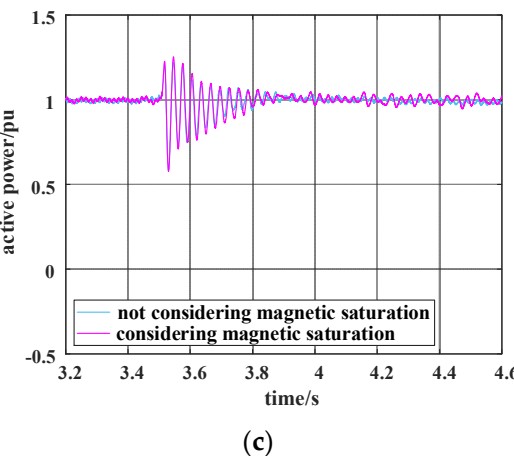

(**c**)

**Figure 17.** Comparison of the system active power under different $K_{p1}$ with and without magnetic saturation: (**a**) when $K_{p1}$ = 0.8, comparison of system active power with and without magnetic saturation; (**b**) when $K_{p1}$ = 2, comparison of system active power with and without magnetic saturation; (**c**) when $K_{p1}$ = 2, comparison of subsynchronous suppression of the system with and without considering magnetic saturation.

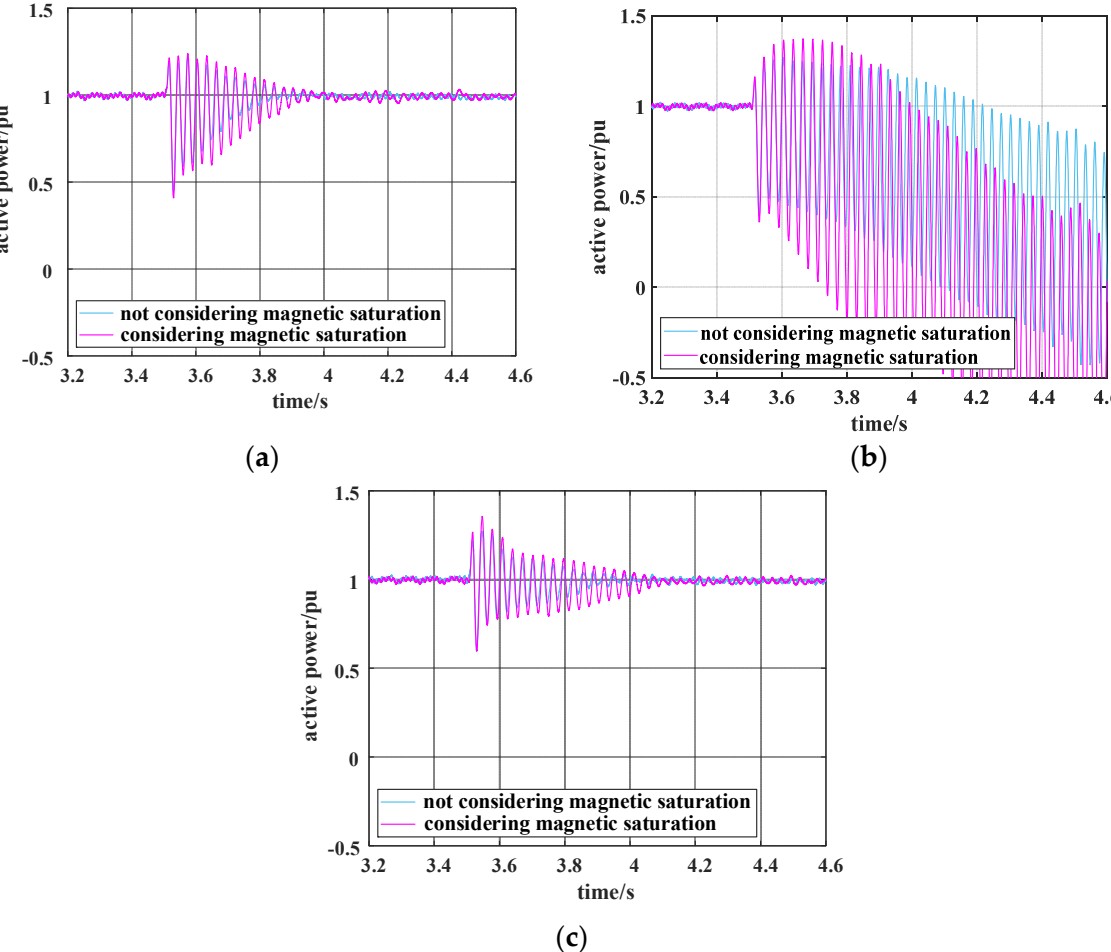

**Figure 18.** Comparison of the system active power with and without magnetic saturation under different $K_{i1}$: (**a**) when $K_{i1}$ = 0.1, comparison of system active power with and without magnetic saturation; (**b**) when $K_{i1}$ = 0.007, comparison of system active power with and without magnetic saturation; (**c**) when $K_{i1}$ = 0.007, comparison of subsynchronous suppression of the system with and without considering magnetic saturation.

## 6. Conclusions

In this study, an equivalent impedance model considering RSC and DFIG was developed. In addition, the affecting factors and mechanisms for SSO in the system were analyzed in combination with the equivalent RLC resonant circuit of the DFIG-based series-compensated grid-connected system. The SSO suppression was transferred to the control of the converter, and the system stability was improved by eliminating the SSO by attaching a virtual resistance controller to RSC. Overall, the following conclusions are drawn by simulation verification:

- Larger SCD and proportionality coefficient values and a smaller integration coefficient value lead to a greater susceptibility to SSO;
- SSO can be suppressed by parameters optimization. SCDs of line should be minimized without reducing the transmission capability. Additionally, the proportionality coefficient $K_{p1}$ can be reduced and the integration coefficient $K_{i1}$ increased to enhance system stability.

**Author Contributions:** This paper was completed by the authors in cooperation. G.G. carried out theoretical research, data analysis, simulation analysis and paper writing. Y.L. and X.W. provided constructive suggestions. H.W. and L.W. revised the paper. All authors have read and agreed to the published version of the manuscript.

**Funding:** This work was supported by Development and Industrialization of Intelligent Wind Farm Holographic State Accurate Perception and Optimization Decision System Project (2021JH1/10400009) and Liaoning Provincial Central Government Guides Local Science and Technology Development Fund Projects (2021JH6/10500166).

**Institutional Review Board Statement:** Not applicable.

**Informed Consent Statement:** Not applicable.

**Data Availability Statement:** Data sharing not applicable.

**Conflicts of Interest:** The authors declare no conflict of interest.

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
