# Peer review of "Sub-Synchronous Oscillation Suppression Strategy Based on Impedance Modeling by Attaching Virtual Resistance Controllers for Doubly-Fed Induction Generator"

_electronics, doi:10.3390/electronics11142272_

Round 1
Reviewer 1 Report
This paper deals with the interesting problem of oscillation suppression of a doubly fed induction machine.
Below are the comments:
1) In Chapter 2, the authors assumed that a long line with a voltage of 330 kV is analyzed. In such cases, we are dealing with a transmission line with distributed rather than lumped parameters (the model adopted by the authors under such assumptions is incorrect). Additionally, for 330 kV lines, it is advisable to take into account the ground capacitance of the transmission line. The values of line parameters can be found, for example, in books of power system stability studies.
2) The mathematical model of the induction machine adopted by the authors for this study is very simplified. It does not take into account the phenomenon of core saturation. Such models are nowadays already a standard. Therefore, it is advisable to adopt a model that at least in an approximate way takes into account this phenomenon.
3) What is the value of the capacitance of Cg? It is useful to give this value by citing the literature.
4) The classical Fourier transform FFT is dedicated to the calculation of the spectrum of steady-state waveforms. The authors calculate the FFT for transient waveforms? This issue needs to be clarified in the paper.
5) The article is worth supplementing with experimental results. Only the results of simulation studies are included in the article.
6) How were the parameters of the control system Kp1 Ki1 shown in Fig. 3 selected ? Selection of optimal parameters of the regulator can be searched for using, for example, the method of least squares. For such selected parameters of the control system it is worth to present the results of calculations.
7) The text of the article from the fourth page is formatted in a different way (there are more spaces between lines).
Reviewer 2 Report
- ωe should be explained from equation (4) to equation (8) .
- ZRSC should be defined in equation (11).
- The nunberical vaules of Rsum, Lsum and Cg in Figure 4 should be listed.
- Some parameters should be explained in details.
Round 2
Reviewer 1 Report
The revised version of the article contains all the improvements that were included in the first round of reviews. The authors' responses are, in my opinion, comprehensive. In view of the above, I recommend the article for publication in the mdpi journal.
Comments: The article still requires editorial correction (the text interlines are different on pages 1-3 and different from page 4-21 of the article). In line 318 there is "there..." and should be "There...". The formatting of the literature references still needs minor improvement.
Author Response
请参阅附件。

Reviewer 2 Report
This paper has been revised by the Authors according to the reviewer's comments, This paper should been accepted.
Author Response
The authors thank the honourable reviewer for the wonderful review and insightful comments on the paper, as these comments have promoted the improvement of the work.